# Compact Camera Fluorescence Detector for Parallel-Light Lens-Based Real-Time PCR System [note 1]

**DOI:** 10.3390/s22218575

**Published:** 2022-11-07

**Authors:** Seul-Bit-Na Koo, Yu-Seop Kim, Chan-Young Park, Deuk-Ju Lee

**Affiliations:** 1School of Software, Hallym University, Chuncheon-si 24252, Korea; 2Bio-IT Research Center, Hallym University, Chuncheon-si 24252, Korea

**Keywords:** real-time PCR, Fresnel lens, fluorescence detection, open platform camera, image processing

## Abstract

The polymerase chain reaction is an important technique in biological research. However, it is time consuming and has a number of disadvantages. Therefore, real-time PCR technology that can be used in real-time monitoring has emerged, and many studies are being conducted regarding its use. Real-time PCR requires many optical components and imaging devices such as expensive, high-performance cameras. Therefore, its cost and assembly process are limitations to its use. Currently, due to the development of smart camera devices, small, inexpensive cameras and various lenses are being developed. In this paper, we present a Compact Camera Fluorescence Detector for use in parallel-light lens-based real-time PCR devices. The proposed system has a simple optical structure, the system cost can be reduced, and the size can be miniaturized. This system only incorporates Fresnel lenses without additional optics in order for the same field of view to be achieved for 25 tubes. In the center of the Fresnel lens, one LED and a complementary metal-oxide semiconductor camera were placed in directions that were as similar as possible. In addition, to achieve the accurate analysis of the results, image processing was used to correct them. As a result of an experiment using a reference fluorescent substance and double-distilled water, it was confirmed that stable fluorescence detection was possible.

## 1. Introduction

In terms of worldwide overall mortality, infectious diseases are the leading cause of death. The prevalence of infectious diseases has declined due to vaccination and the development of antibiotics, but they are still considered a significant problem. Despite the development of medicine, infectious diseases continue to change and emerge for various reasons, such as environmental changes, social factors, and pathogenic adaptation and change [1,2,3,4,5,6,7].

In recent years, the prevalence of new infectious diseases has been increasing rapidly, and they have been appearing in shorter cycles. These diseases are mostly caused by wild animals, and in time, they can turn into large-scale epidemics, affecting public health and the social economy [2,6,7,8]. Most infectious diseases are transmitted through the respiratory tract, but the recent COVID-19 virus has a novel transmission pathway by being transmitted through the respiratory tract and surviving and transmitting on objects and surfaces. As a result, it is highly contagious. It is difficult to control the spread of infectious diseases, leading to economic instability worldwide [6,7].

The dangers of infectious diseases include high mortality, drug development and approval later than a disease’s emergence, and virus resistance and evolution. To suppress infectious diseases, the early diagnosis of infected individuals is important. An accurate and quick diagnosis can save a patient’s life and prevent the spread of infection in their community. Therefore, technology that enables rapid and accurate detection and analysis plays an important role in the response and prevention of infections and infectious diseases [8,9,10,11,12,13]. Tests and diagnoses for most infectious diseases use culture-based serological analysis as a standard method. However, this method is labor-intensive and time consuming, and it could provide inaccurate results. The molecular diagnostic test method based on a polymerase chain reaction has excellent reproducibility and sensitivity, and it does not take as long to perform this test [8,9,14,15]. Detection through a polymerase chain reaction, which is the basis of molecular diagnosis, involves several steps and requires special equipment and skilled professionals. A real-time polymerase chain reaction device that integrates the amplification and detection of nucleic acids has been developed to solve this problem. This device does not transfer samples to a post-processing device in the detection step and can detect a wide range of pathogens [14,15,16,17,18].

Recently, many studies have been conducted with the aim of shortening the inspection time and improving diagnostic ability with regard to infectious diseases, and field tests are also actively being carried out. As these viruses mainly occur in developing countries and rural areas, POC (point-of-care) applications are needed in areas with insufficient medical facilities [11,12,13,19].

A real-time polymerase chain reaction consists of a temperature controller for DNA amplification, a fluorescence detection system, a controller, and a graphic display. In general commercial products, the fluorescence detection system has a complex configuration, including a high-performance image detector and various optical components. In an imaging device for fluorescence detection, an expensive camera and a photodiode are generally used. However, an optical distance is required to detect fluorescence, which is bulky and expensive [8,14,15,20,21,22,23,24].

Recently, many small cameras with good performance have been developed due to the continuous development of cameras in smartphones. Accordingly, a small camera with excellent performance and a lowered manufacturing cost is being developed. In addition, many open platforms have been developed for image analysis, making it easy to analyze fluorescence images [17,20,21,22,23].

A real-time polymerase chain reaction device detects many samples at once. When analyzing the fluorescence of a sample, it is most important to obtain accurate experimental results by shooting under the same conditions. In a fluorescence detector, additional components, such as optical lenses, are used to improve the detection performance [23,24]. Image lenses, such as telecentric lenses, can present problems due to their complicated design method and lens size according to the FOV. In addition, when using these optical lenses, it is difficult to collimate the light when the light source is far away, and they are expensive.

In this paper, we propose the use of a low-cost, compact fluorescence detection device using a small and inexpensive open platform COMS camera and a Fresnel lens. The basic concept of the proposed system was presented at a conference, and this paper is an extended version of the content presented there [24]. We used a small CMOS camera instead of an existing expensive camera to lower the cost and reduce the size of the camera. The case was manufactured using matte acrylic to block external light. The camera and LED were placed close to the center of the Fresnel lens. As the lens of the compact CMOS camera is small, a detector for multiple detections was manufactured by attaching the light source as close as possible on the same line. We used a Fresnel lens with a concentric groove on the lens surface to detect 25 tubes with one light source and camera to eliminate parallax errors. The system’s control unit was configured to control several local systems connected by a single USB (universal serial bus) from the host computer.

To verify the system proposed in this paper, a comparison experiment was conducted by measuring the brightness using a fluorescent dye reagent (FAM) and double-distilled water (DDW). The verified experimental results were obtained through qualitative and quantitative analyses. Although the proposed system has a simple structure and can be created at a low cost compared to existing commercial devices, our results show that it is potentially a suitable device for use in fluorescence detection.

## 2. Materials and Methods

Figure 1a shows the overall structure of the system we proposed in this paper. The structure of the system was largely divided into a detection component and an amplification component and was configured in a vertical form. In the amplification component, a thermocycler owned by this research team was used. The amplification component was divided into a lid heater, used to heat lids; a heating chamber, used to control the temperature of a heating block in which a tube containing reagents can be placed; and a main board, used to control the entire system. The lid heater suppressed evaporation by heating the tube cover to a high temperature to prevent the evaporation of the sample as it was heated to a high temperature in the thermal denaturation part of the PCR process. The lid heater included an aluminum plate and a flexible PCB, and the temperature was measured using a connected thermistor. The heating chamber consisted of a heating block, Peltier, heat sink, and fan. The heating block, which was a space in which the tube containing the reagent could be placed, was made of aluminum with high thermal conductivity, and each well into which the tube was inserted was designed to increase adhesion. A Peltier was attached to the bottom of the heating block to control the temperature of the chamber. A thermistor was attached to measure the temperature between the heating block and the Peltier, and a heat sink was installed to dissipate the heat in the Peltier. A space for a thermistor to be mounted was left between the Peltier and the heat sink to measure the temperature of the heat sink and thus control the speed of the fan.

In order to measure the fluorescence, one open platform CMOS camera with a small aperture was fixed in the center, and the optical part had a simple design. One white LED light source was placed close to the camera and fixed. For multiple fluorescence measurement, four excitation filters and four emission filters in 5 dia size were arranged in parallel with the LED and manufactured as one module. The excitation filter was placed in front of the LED, and the emission filter was placed in front of the camera to detect the fluorescence reflected by the tube. By placing a Fresnel lens on the reed heater, a total of 25 tubes were designed so that a single camera with a short focal length could capture them uniformly.

Figure 1b shows the detailed function of the entire system through a block diagram. The microcontroller and camera were individually connected to a commercial USB 2.0 hub to form a single local host system. The heating cycle of the PCR was executed by measuring and controlling the temperature using the thermistor attached to the lid heater and the Peltier using a microcontroller. In addition, the microcontroller used an Arduino to control several peripherals. The LED brightness was controlled using PWM (pulse width modulation) through the LED driver, and the linear stepper motor was controlled using the motor driver. The starting position of the motor was set through the sensor, and the filter module, including the excitation filter and emission filter in four pairs, moved to match the LED and the camera. For the initial experiment, in the detection component, which included the lid heater, a linear actuator was used to set the height between the tube and the detection component before the experiment. The camera we used includes a Sony imx298 sensor used in smart device imaging devices, and USB communication is possible with this device.

Figure 2a shows the system used in the actual experiment, except for the thermocycler. Figure 2b shows a system with a linear actuator in the initial setup. In order to prevent light reflection and light penetrating from outside, a dark room was created by assembling a matte black acrylic wall. The system was also secured using matte black aluminum supports. At the bottom of the system, there was a heating block with 25 wells into which 25 tubes could be inserted. For the lid heater, 1 mm of black aluminum, which was as thin as possible, was used, and 25 holes were drilled to capture the fluorescence of the tube. The Fresnel lens was effective when its diameter was larger than the diameter of the heating block, and a lens similar to the focal length of the camera was selected. The lid heater and Fresnel lens had to be mounted at a minimum distance. In addition, during the continuous PCR experiments, there was a possibility that deformation would occur due to the heat of the lid heater, so a distance of 5 mm was used.

The detection component consisted of a filter wheel board including a filter, an LED, a linear stepper motor, and a camera board with a camera. The distance between the Fresnel lens and the filter wheel board was set to 70 mm, which was the effective focal length of the Fresnel lens. All optical components used in the system were manufactured to be matte black, except for the heating block, to compensate for the SNR (signal-to-noise ratio). Table 1 shows the main components used in the experiment, and Table 2 shows the specifications of the Fresnel lens.

The proposed system was designed to enable multiple fluorescence detection. The types of detectable fluorescence are fluorescein (FAM), hexachloro-6-carboxyfluorescein (HEX), 6-carboxyl-X-rhodamine (ROX), and cyanine 5 (CY5). Therefore, in order to detect four types of fluorescence, four excitation filters and four emission filters were required. As fluorescent light is irradiated as parallel light through a Fresnel lens, the angles of the excitation filter and the emission filter were not adjusted and were placed side-by-side on a plane.

Figure 3a,b show the back and front of the system board to which the filter wheel board was attached. A filter wheel was manufactured for use in multiple fluorescence detection, and a 2 mm-sized LED was adopted to irradiate light onto DNA containing fluorescent substances. The LED was attached to the PCB (printed circuit board) and was attached in the same direction as the excitation filter on the aluminum plate of the filter wheel board.

A thermal pad was attached between the PCB and the aluminum to which the LED was attached to prevent problems caused by heat generated by repeated LED use. The camera used in fluorescence detection was fixed in the same direction as the emission filter. As the lens diameter and aperture became smaller by choosing a smaller LED and a mini smartphone camera with a small sensor, the excitation and emission filters could be used as small filters with diameters of 5 dia and thicknesses of 1 mm. Therefore, the distance between the emission filter and the excitation filter could be narrowed so that it would be as close to the center of the Fresnel lens as possible.

The filter wheel had eight holes with sizes of 5 mm, and there was a 3 mm magnetic hole used to sense the filter groove. At the bottom of the filter wheel, 0.5 mm of thin aluminum was added to prevent the filter from falling down. The excitation filter did not need a cover because the board to which the LED was attached was covered. In the case of the emission filter, a finger board was required to connect the step motor and allow the filter wheel to move. Therefore, it was manufactured as a step-type filter wheel, and sliding Teflon tape was attached to the frictional surface to enable smooth movement. A filter wheel holder was used to block external light and support the weight of the filter wheel. The filter wheel board and its components were matte black to prevent light leakage. The camera was mounted in the same position as the emission filter.

Table 3 shows the images and specifications of the Sony IMX298 imaging system. The open platform CMOS camera used in this experiment includes a Sony IMX298 image sensor, and autofocus and USB connection are possible with this device. In addition, it supports a maximum resolution of 4656 × 3496, and in this experiment, we placed it as close to the filter as possible so that it did not block the viewing angle.

After adjusting the excitation light emitted from the LED to be parallel, the LED needed to be placed at an effective focal length height to emit the fluorescence reacted by irradiating the lid of the tube containing the DNA. In addition, it had to be placed at the effective focal length height of the Fresnel lens to prevent side distortion when 25 tubes were photographed with one camera. It was impossible to center the LED and the camera at the same time on the Fresnel lens. Therefore, an experiment was conducted to determine the best position of the camera and the LED from the focal length of the Fresnel lens to obtain the maximum-performance fluorescence brightness. The minimum distance between the LED and the camera was used to prevent light interference.

Figure 4 shows four experimental methods for the detection of stable fluorescence. To achieve the effect of a Fresnel lens, the camera and LED were located in the center of the lens at the effective focal length of the lens. In order to obtain the maximum SNR, an experiment was conducted according to the position of the camera and LED. In the four figures in Figure 4, the large circles show the Fresnel lens, and the 5 × 5 blue circles inside the square plate represent the tubes containing the reagents. The orange dot represents the camera and LED corresponding to the center of the tube and lens located at (3.3) on the plate. The orange line shows the orientation of the camera and lens from the center. Figure 4a,b show the experimental methods used to investigate the change in fluorescence brightness according to the arrangement of the camera and LED. Figure 4a shows how the camera was positioned in the center of the Fresnel lens and position the LEDs towards the second quadrant. Figure 4b shows how to position the LED in the center of the Fresnel lens and position the camera towards the second quadrant. Figure 4c,d show the experimental method used to obtain the change in brightness according to the position movement after positioning the camera and LED. In both experiments, the camera board and the filter wheel board were fixed. Figure 4c shows the method of arbitrary rotation from 1 mm to 6 mm in 1 mm increments to detect the change in fluorescence. The support located in the third quadrant (gray dot) was fixed, and the support in the first quadrant (blue dot) was rotated in the direction of the second quadrant. The supports were spaced and collinear with the center of the tube at each corner. After using the method in Figure 4c, the camera and LED were positioned in the same radial direction from the center of the Fresnel lens to reduce light reflection. Figure 4d shows how an ideal fluorescence value can be obtained depending on how far away it is in the 45-degree direction from the center of the Fresnel lens.

A comparative experiment to verify the system was performed to compare the difference in brightness values using a reference fluorescence reagent (FAM) with a fluorescence concentration when the DNA was maximally amplified and double-distilled water was used (DDW). A total of 25 tubes were composed of a 5 × 5 type FAM plate and a DDW plate so that they could be placed in a 5 × 5 well heating block. In the experiments shown in Figure 4a,b, the brightness of the fluorescence was obtained via photography without positional movement. In the experiments shown in Figure 4c,d, the plate was rotated 180 degrees for accuracy and consistency, and the fluorescence brightness was compared after additional imaging. The images of the FAM plate taken without positional movement are denoted by f0, and the images of the DDW plate are denoted by d0. The images in which the f0 plate is rotated 180 degrees are denoted by f1, and the images in which the d0 plate is rotated 180 degrees are denoted by d1. Therefore, in Figure 4c,d, the FAM plate images (f0,f1) and DDW plate images (d0,d1) were obtained. For each plate image combination, (d0,f0), (d0,f1), (d1,f0), and (d1,f1), fluorescence analysis was performed to determine the uniformity of the plate space.

In order to obtain the brightness from the photographed fluorescence image, it was necessary to obtain a fluorescence range. In an actual real-time PCR device, fluorescence images are not shaken by inserting the tube into a fixed heating block. In this experiment, the chamber may have been shaken due to physical phenomena such as changes in the position of the tube in the heating block by rotating a 5 × 5 plate consisting of 25 tubes, or when the FAM plate and DDW plate were replaced in the heating block. Therefore, image processing was used for the captured images.

Figure 5a shows the image-processing process for the area of the tube where fluorescence is visible after obtaining the entire image of 25 tubes. In the DDW plate image (d0,d1), there are cases where fluorescence cannot be seen even if the brightness is increased to increase visibility. Therefore, images from the FAM plate were used to determine the properties of the tube. The FAM plate images (f0) taken in all four of the experiments were obtained using an image-processing program (ImageJ). Information regarding the center and diameter of the tubes located at (1.1), (1.5), (5.1), and (5.5) of the plate was directly marked and obtained. As the distance between the wells was constant, the spacing of the tubes in the plate was the same. Therefore, information regarding the tube at each corner and the bilinear interpolation method could be used to obtain information regarding the centroids of 25 tubes in plate f0. The centroids of the 25 tubes obtained through plate f0 were substituted for plate images f1, d0, and d1. The brightness of the fluorescence was detected by designating a region of interest equal to the size of the radius from the center of each tube. After determining the radius at which the most stable brightness was obtained during fluorescence detection according to the radius, the brightness within the corresponding range was compared. Figure 5b shows an image of the plate taken after being pushed 4.5 mm in a 45-degree radial direction from the center during the experiment shown in Figure 4c. In Figure 5b, the upper right image is an ROI image of the unrotated plate image f0. In Figure 5b, the lower right image is a d0 image obtained by ROI by substituting the information regarding the center of the tube obtained through f0. The two instances of fluorescence in the center of f0 and d0 were not relevant as they were light reflected by the LED. The red dot represents the center of each tube, and the blue circle represents the actual size when set to a radius of 50 pixels.

## 3. Results

Figure 6 shows (a) the FAM plate f0 image and (b) DDW plate d0 image from an experiment that investigated the fluorescence brightness when the camera was placed in the center of the Fresnel lens and the LED was placed on the outside. The camera was set to gain 0, PWM 200. In the FAM plate in Figure 6a, the (4.5) and (5.5) position tubes were injected with DDW to determine the presence or absence of FAM reagent. In Figure 6a, the dots surrounded by the red circles around (3.3) are the light reflected by the Fresnel lens, and the small dots surrounded by the yellow circles are the light reflected back through the Fresnel lens. Therefore, the DDW plate image d0 in Figure 6b also displays reflections. In addition, the d0 plate image shows intermittent fluorescence reflection in the second and fourth quadrants.

Figure 7a,b show the mean brightness of each of the 25 tubes and the difference between the mean brightness of the tubes in the same position. The blue scatter in Figure 7a indicates f0, and the yellow scatter indicates d0. The *x*-axis in Figure 7b represents a column number of tubes, and each scatter shows a row of tubes for each column. Referring to Figure 7a (f0), the brightness of the upper and lower tubes at both ends appeared to drop, and in the scatter in Figure 7a (d0), it can be seen that abnormally high fluorescence brightness appeared due to reflection. Therefore, in the f0 image in Figure 7a, the fluorescence brightness of the tube (20.25) injected with DDW reagent was scatter 4 and 5 on the *x*-axis 5 of Figure 7b. In the d0 plate image in Figure 6b, the tubes (tube index 6 in Figure 7a) with high brightness due to reflection were the second scatter of *x*-axis 1 (column number) with a small difference in brightness in Figure 7b. For 22 tubes, the mean brightness of f0 was over 140, and the mean brightness of d0 was about 33—this excluded the tubes with the three prominent fluorescence values mentioned above. The mean of the relative brightness gain of the 22 tubes at the same position on the f0 plate and the d0 plate was 3.5, and the relative standard deviation of the mean brightness difference was 13.9.

Figure 8 shows a d0 image when the LED was placed in the middle of the Fresnel lens and the camera was placed on the outside. The camera was set to gain 0, PWM 200. The distance between the camera and the LED was set to be the same. The d0 image shows that the fluorescence brightness of the tubes was not constant, and the light reflection was severe.

Figure 9a shows the mean brightness of f0 and d0, and Figure 9b shows the difference between the mean brightness of f0 and d0. In the plot of Figure 9a, the values of the mean brightness of f0 and d0 significantly increased when compared to the plot of Figure 7a, in which the camera was placed in the center of the Fresnel lens. The yellow scatter showed a large difference in the brightness gap within the d0. Additionally, the brightness of d0 was similar to that of f0. Figure 9b shows that the brightness values were not similar, and the mean brightness difference was not constant. Thus, it can be seen that both the rows and columns of the tube had dissimilar brightness values. These results indicate that centering the camera in the center of the Fresnel lens obtained more stable results than placing the LED in the center.

Figure 10 shows the d0 image when the filter wheel board and camera board rotated together toward the second quadrant after the camera was placed in the center of the Fresnel lens. The camera was set to gain 0, PWM 200. Figure 10a shows the fluorescence brightness image of the DDW-filled tubes, which were rotated by 4 mm; Figure 10b, 5 mm; and Figure 10c, 6 mm. As the rotation distances increased, almost no reflection was seen in the second and third quadrants.

The plots in Figure 11 show the minimum difference in brightness collected by widening the fluorescence range radius at the center of each of the 25 tubes in the FAM plate (f0,f1) and DDW plate (d0,d1). For accuracy, two images by plate were repeatedly crossed and analyzed. Therefore, there are four graphs per plot. The *x*-axis shows the fluorescence range radius and the *y*-axis shows the fluorescence intensity value. The smaller the fluorescence range radius, the larger the value of the minimum difference in brightness value. As shown in Figure 10, the four graphs for the crossed images are stably displayed with increasing rotation distance. In addition, the most stable graph between a radius of 40 and 100 is shown.

Figure 12a,b show the mean brightness and mean brightness difference of the FAM plate (f0,f1) and DDW plate (d0,d1) image combinations when the filter wheel board and camera board were rotated by 6 mm. In the plot of Figure 12a, the shape of the scatter flow of the DDW plate d0,d1 is stable, and the FAM plate f0,f1 is not as constant as the DDW plate, but it shows that they are close to each other. In the FAM plate, the DDW reagent was injected into (4,5), (5,5) from the forward direction for identification. Therefore, the fluorescence brightness at indexes 20 and 25 of f0 and indexes 1 and 6 of f1 decreased. However, the overall fluorescence intensity values of f0 and f1 indicate that they were not as bright as when the DNA was amplified. The plot of Figure 12b indicates that blue and green scatterers combining f0 images on *x*-axis 20 and 25, and orange and red scatterers combining f1 images on *x*-axis 1 and 6, were tubes in which DDW reagent was injected into the FAM plate. Except for the scatterers mentioned, the difference in the mean values was remarkably stable. When the camera and the LED were placed in the same radial direction from the center of the Fresnel lens, the reflection was reduced and the fluorescence could be reliably detected. If the fixed point, the movement start point, and the movement point were expressed as a line segment, the length of the rotation distance was 6 mm and the camera placed in the center of the Fresnel lens was the midpoint of the line segment. The moving distance of the camera could be expected to be about 3 mm by the double-point theorem.

Figure 13 shows the d0 image shifted 45 degrees from the center of the Fresnel lens (a -> b -> c -> d -> e). In the rotation experiment, good results were obtained at a distance about 3 mm away from the center of the lens, so images from 3 mm and 4 mm to 5.5 mm were taken at 0.5 mm intervals. The camera was set to gain 0, PWM 200.

The plots in Figure 14 show the minimum difference in luminance collected by widening the fluorescence range radius at the center of each of the 25 tubes containing f0,f1 and d0,d1. The figure shows a stable graph when it is offset by about 4.5 mm. It can also be seen that there is no difference in brightness between 50 pixels and 100 pixels for changes in image composition and radius.

Figure 15a,b show the mean brightness and the difference between the mean brightness according to the combination of images taken by taking f0,f1 and d0,d1 when the camera and LED were offset 4.5 mm from the center of the Fresnel lens. The size of the fluorescence range radius in Figure 15a was set to 50 pixels. The brightness of f0,f1 showed a constant shape within 230 (DNA saturation value 255), and in the case of d0,d1, the change in brightness was similar in each row of the tube. In terms of the scatter in Figure 15b, there was little change in the brightness of the tube in each location.

Table 4 shows statistics based on the average brightness within a radius of 50 pixels and 100 pixels. Using f0,f1 and d0,d1, this was calculated through the mean difference of a total of four combinations. The spatial uniformity can be seen in the combination of the four images. Rel.max means the displacement coefficient, and the smaller the value, the more gathered the distribution is, so the actual performance effect can be seen. The SNR decreased as the difference in brightness between the FAM plate and DDW plate increased. As Min is the minimum value of the brightness difference, the larger the value, the lower the SNR. In addition, the smaller the minimum Max/Min, the better the performance. In terms of Min, 4.5 mm (50 pixels) showed a value of 136.316. In Max/Min, 4.5 mm (50 pixels) showed a value of 1.271429, and Rel.max showed a value of 7.913383 at 5 mm.

## 4. Discussion

In this paper, we proposed the use of a compact camera fluorescence detection device for real-time PCR based on a parallel-light lens, which can be implemented at a low cost using an open platform camera and a Fresnel lens. A detection component for fluorescence measurement was simply designed by reducing the optical components using a smartphone camera. In addition, a Fresnel lens was used to capture a large detection area with a camera with a short focal length. The host local system was used to efficiently run the function. To verify the fluorescence measurement performance, it was confirmed that stable fluorescence detection was possible through experiments and analysis using FAM and double-distilled water. In the experiment that assessed the change in brightness according to the arrangement of the camera and the light source in the center of the Fresnel lens, it was shown that the brightness of the tubes was stable when the camera was placed in the center of the Fresnel lens. However, there was intermittent reflection. When the camera was placed in the center of the Fresnel lens, and the camera and the light source were rotated in one direction, and it was found that the reflection disappeared visually when the camera moved by about 6 mm. It was found that the random reflection of fluorescence could be reduced by positioning the camera and light source in a radial direction. In order to detect the most ideal fluorescence brightness, the changes that occurred when the camera and light source (LED) were placed away from the center of the Fresnel lens were observed. In addition, spatial uniformity was investigated through statistics according to the average brightness within the radius of 50 and 100 pixels. At 4.5 mm (50 pixels), the minimum value of the mean was the highest at 136.316, and the maximum/minimum value of the mean was 1.271429, which converged closest to 1. Additionally, the rel.max was 7.913383, which was the closest to 0. In the future, for more accurate fluorescence measurement, the intermittent micro-movement of the module will be resolved through the use of a precise system structure. Furthermore, we plan to conduct comparative experiments with commercial devices by supplementing and completing a prototype that can be used for actual DNA amplification experiments, which are currently in progress.

## Figures and Tables

**Figure 1 sensors-22-08575-f001:**
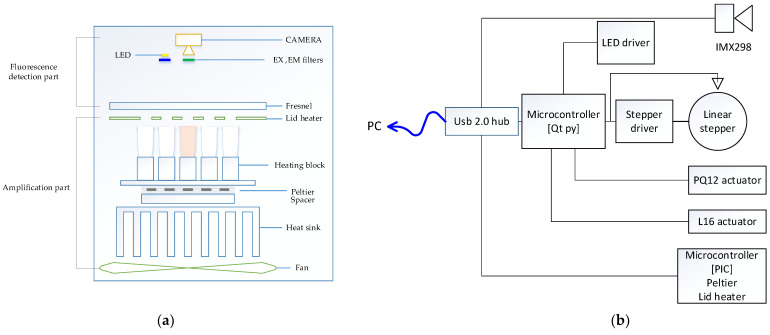
(**a**) System structure diagram; (**b**) system block diagram.

**Figure 2 sensors-22-08575-f002:**
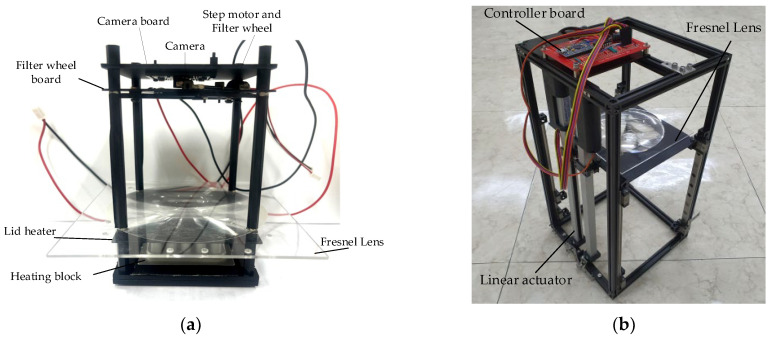
(**a**) System used in the actual experiment, except for the thermocycler; (**b**) linear actuator for initial height setting.

**Figure 3 sensors-22-08575-f003:**
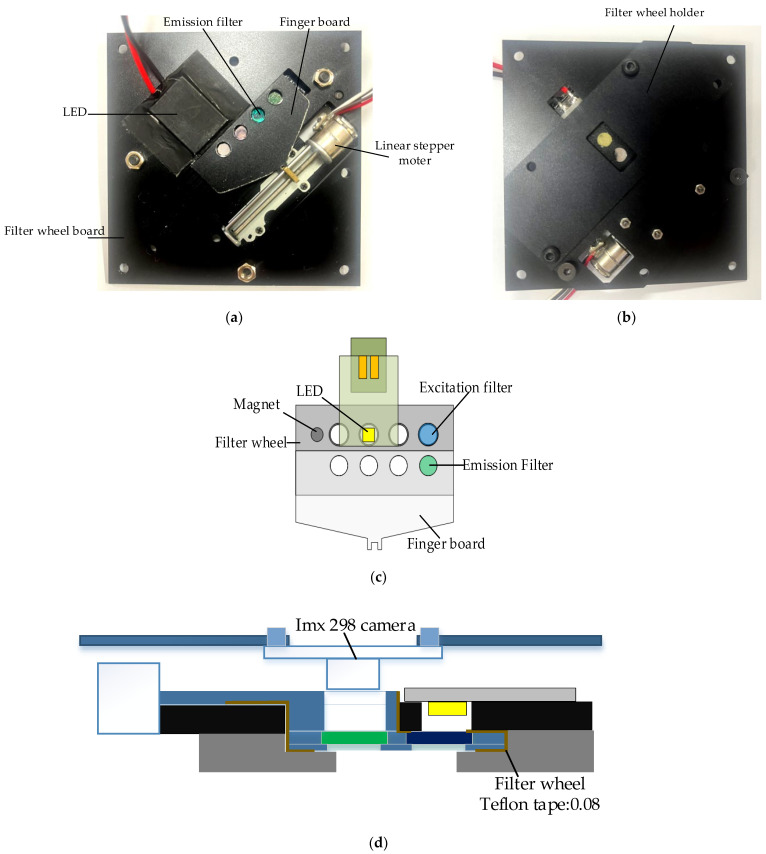
(**a**) Back of system board with filter wheel module and LEDs attached; (**b**) front of system board with filter wheel module attached; (**c**) system structure diagram of filter wheel module; (**d**) side view of filter wheel module and detector module.

**Figure 4 sensors-22-08575-f004:**
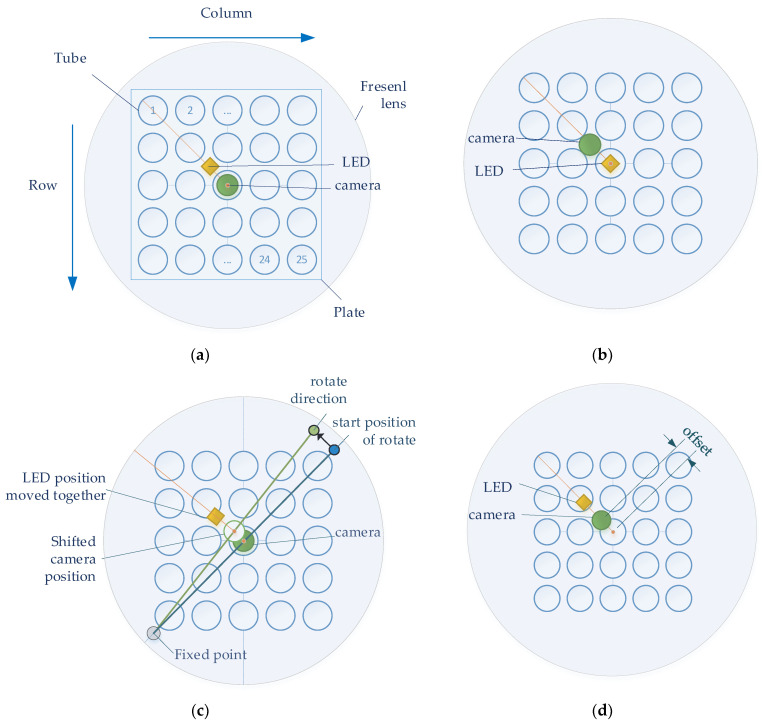
(**a**) Center the camera; (**b**) center the LED; (**c**) rotate the camera and LED in the same direction; (**d**) move the camera and LED in a 45-degree direction.

**Figure 5 sensors-22-08575-f005:**
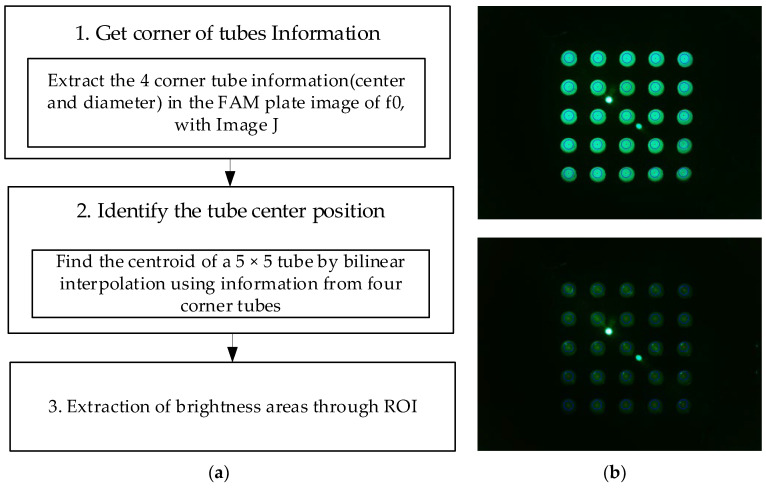
(**a**) Image processing for 25 tubes; (**b**) FAM plate (f0) and DDW plate (d0) images.

**Figure 6 sensors-22-08575-f006:**
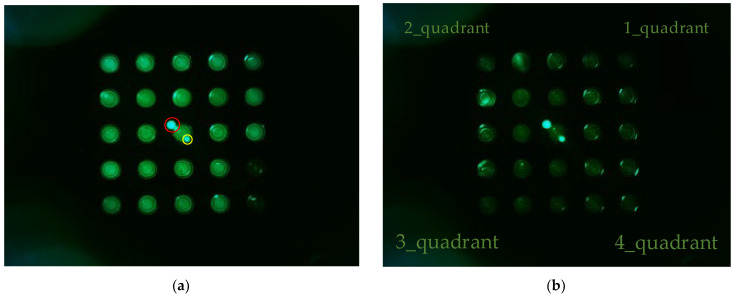
(**a**) FAM plate image f0; (**b**) DDW plate image d0.

**Figure 7 sensors-22-08575-f007:**
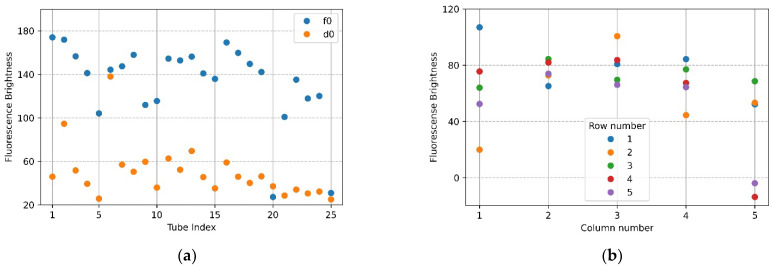
Brightness comparison for f0 FAM plate image and d0 DDW plate image taken with the camera placed in the center position of the Fresnel lens: (**a**) mean brightness of f0 and d0; (**b**) difference between the mean brightness of f0 and d0.

**Figure 8 sensors-22-08575-f008:**
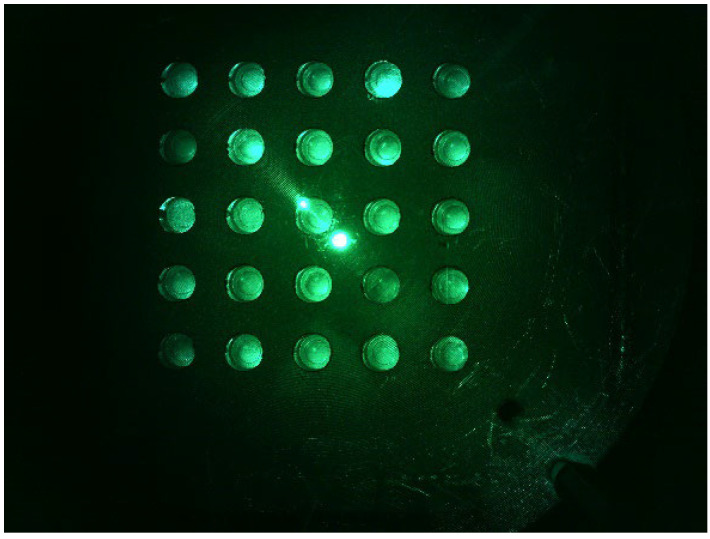
d0 DDW plate image with the LED centered and the camera positioned sideways.

**Figure 9 sensors-22-08575-f009:**
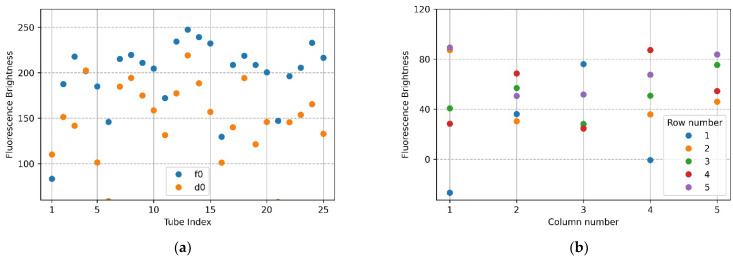
(**a**) Brightness comparison plot for f0 FAM plate image and d0 DDW plate image taken with the LED placed in the center position of the Fresnel lens: (**a**) mean brightness of f0 and d0; (**b**) difference between the mean brightness of f0 and d0.

**Figure 10 sensors-22-08575-f010:**
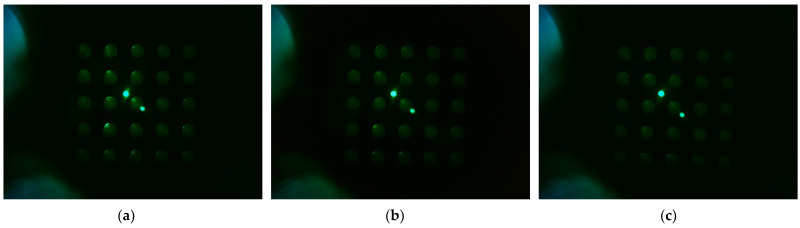
Change in brightness of plate d0 image when rotating in the direction of quadrant 2 with camera placed in the center of lens, LED placed outside, and camera board and filter wheel board fixed to each other: (**a**) brightness when rotated by 4 mm; (**b**) brightness when rotated by 5 mm; (**c**) brightness when rotated by 6 mm.

**Figure 11 sensors-22-08575-f011:**
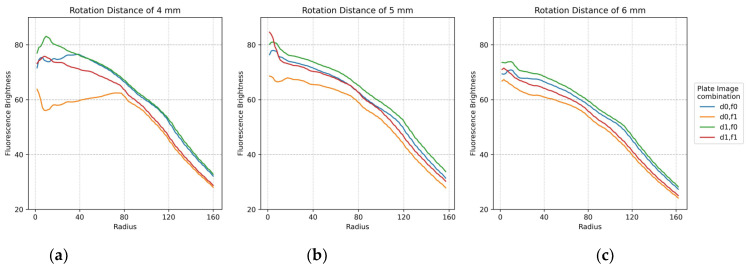
Minimum difference in mean brightness as of fluorescence radius for f0,f1 FAM plate image and d0,d1 DDW plate image combinations according to rotation distance: (**a**) 4 mm; (**b**) 5 mm; (**c**) 6 mm.

**Figure 12 sensors-22-08575-f012:**
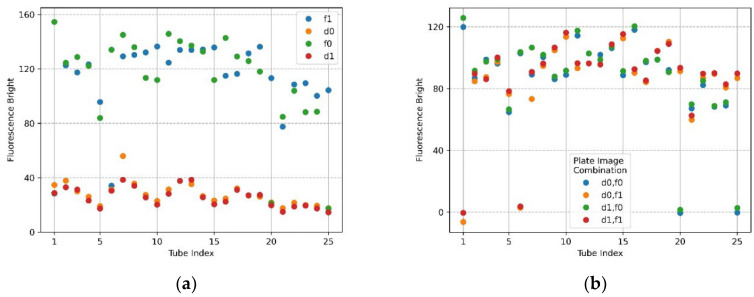
When the camera and LED are rotated 6 mm to the second quadrant, fluorescence brightness as detected by ROI as the radius of 50 for f0,f1 FAM plate image and d0,d1 DDW plate image; (**a**) mean brightness of f0,f1 and d0,d1; (**b**) differences in mean brightness according to FAM and DDW plate image combinations.

**Figure 13 sensors-22-08575-f013:**
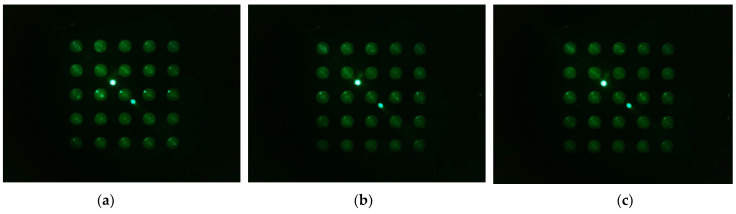
Images of d0 plate taken when the camera and LED, arranged in the radial direction at an angle of 45 degrees in the second quadrant, were offset from the center of the lens by the following distances: (**a**) 3 mm; (**b**) 4 mm; (**c**) 4.5 mm; (**d**) 5 mm; (**e**) 5.5 mm.

**Figure 14 sensors-22-08575-f014:**
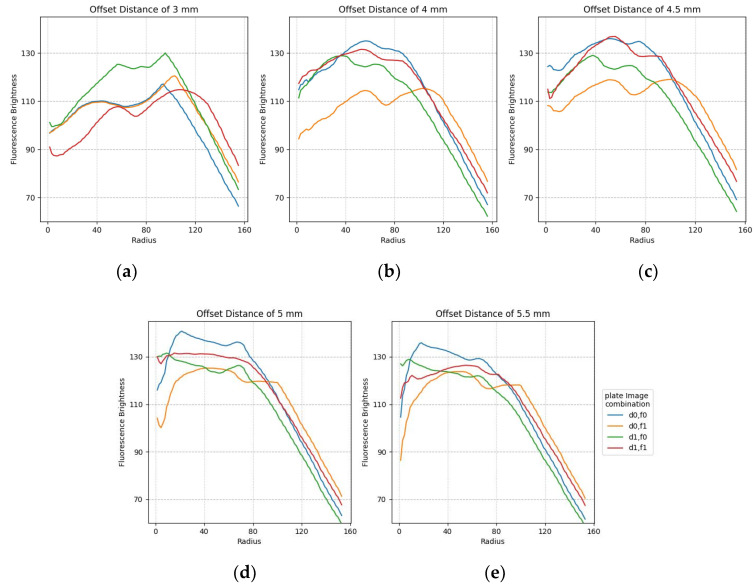
Minimum difference in mean fluorescence brightness according to the radius for FAM and DDW plate image combinations regarding offset distance: (**a**) 3 mm; (**b**) 4 mm; (**c**) 4.5 mm; (**d**) 5 mm; (**e**) 5.5 mm.

**Figure 15 sensors-22-08575-f015:**
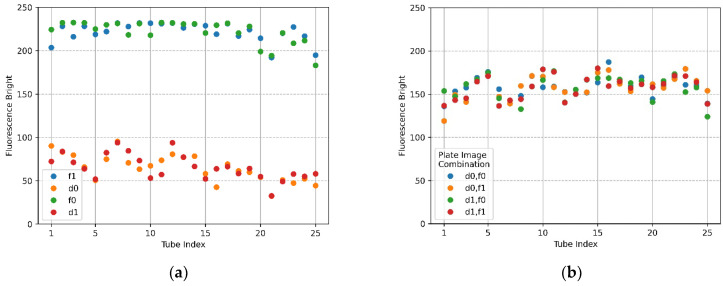
Fluorescence brightness detected as ROI with a radius of 50 for f0,f1 FAM plate images and d0,d1 DDW plate images. The camera and LED arranged in the radial direction at 45 degrees in the second quadrant were offset by 4.5 mm from the center of the Fresnel lens; (**a**) mean brightness of f0, f1, d0, and d1; (**b**) mean brightness differences between FAM and DDW plate image combinations (d0,f0), (d0,f1), (d1,f0), (d1,f1).

**Table 1 sensors-22-08575-t001:** System components.

Classification	Detail	
Imaging system	IMX 298 Sony sensor 60° view angle	
Fluorescence reagent Control reagent	Probe type FAM (0.14 μmol/μL)	
	DDW (double-distilled water)	
Filter	Excitation	CW: 470 nm, BW: 40 nm
	Emission	CW: 535 nm, BW: 50 nm
Light	White LED (350 mA, 140° view-angle)	

**Table 2 sensors-22-08575-t002:** Lens specifications.

Classification	Detail
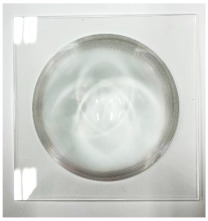	Optic lens: Fresnel lens Dimensions (inches): 5.0 × 5.0 Effective diameter (inches): 4.0 Effective focal length (EFL) (inches): 2.80

**Table 3 sensors-22-08575-t003:** Camera image and specifications.

Classification	Detail
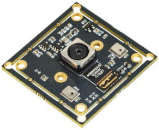	Optical receiver: IMX 298 Focus: Auto Distance: 2 CM–100 M Viewing angle: Autofocus up to 60 degrees

**Table 4 sensors-22-08575-t004:** Image system specifications.

Off Axis (mm)	Rad (Pixels)	Max	Min	Max/Min	Mean	Rel.max
3	50	167.6743	123.8191	1.354188	149.1975	11.83481
	100	163.5046	124.6572	1.311634	148.4835	8.63523
4	50	170.9801	132.705	1.288422	156.6827	10.32022
	100	165.1159	119.1655	1.385602	151.5982	8.823616
4.5	50	173.3161	136.316	1.271429	158.0092	9.020938
	100	166.161	120.9353	1.373966	152.4958	7.275712
5	50	173.2474	134.8673	1.284576	160.3421	8.008093
	100	164.375	112.458	1.461657	150.2266	7.913383
5.5	50	175.928	132.0282	1.332503	161.5325	8.269587
	100	164.3904	110.5677	1.486786	150.9183	8.497039

## Data Availability

Not applicable.

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
