# Peer review of "Compact Camera Fluorescence Detector for Parallel-Light Lens-Based Real-Time PCR System†"

_sensors, 2022, doi:10.3390/s22218575_

Round 1

Reviewer 1 Report

Recommendation: It could be published in Sensors after major revision.

Comments:

The manuscript described a Compact Camera Fluorescence Detector for Parallel Light Lens based Real-Time PCR Devices. The proposed cartridge has simple optical structure and its size can be miniaturized. Authors employed reference fluorescent substance and double distilled water to conform that stable florescence detection with this cartridge was possible. However, the feasibility of the proposed cartridge should be verified by real nucleic acid amplification samples. Moreover, the title of this manuscript is not correct because the proposed cartridge can only be used for end-point detection but not suitable for real-time detection.  Therefore, it is not adequately validated in resent version and should be modified as major revision before published in Sensors.

1. Authors should use real nucleic acid amplification samples to verify the practicability of the proposed cartridge.

2. The title of this manuscript is Compact Camera Fluorescence Detector for Parallel Light Lens based Real-Time PCR Devices. Authors should show the feasibility of the cartridge for real-time nucleic acid amplification and detection.

3. In Figure 2, the name of each element should be marked clearly.

4. In Figure 5, the Image processing algorithm should be optimized. In this version, there is visible florescence in the middle tubes of DDW image. Moreover, the florescence in the image of 25 identical tubes were not balance.

5. Authors should compare the cost of their cartridge with commercial PCR device in details.

6. Can the proposed cartridge make real-time and multiple florescence detection for each sample? The author should detect some real clinical samples to verify the feasibility of the method and make the data more valid.

7. More details of operation should be included in the manuscript. The writing should be further polished in the revised manuscript.

Reviewer 2 Report

-          Figure legends should be more clarified, the authors must add more explanation in the caption of each figure. For example, same caption was used for fig. 7 and 9, there are many colorful dot without any explanation. It is difficult for readerships to observe and understand well the figures.

-          Fig. 13 caption should be shortened by combining the LED (a) 3 mm; (b) 4 mm; (c) 4.5 mm; (d) 5 mm; (e) 5.5 mm apart. Same with other figure captions.

-          The caption of different colorful lines or dots in figure 11, 14, 15 should be revised. X-axis and Y-axis must be shown in the figures. What does “5d0.jpg_5f0.jpg” mean ?

-          Some figures were used with low resolution. Please enhance the resolution for easier observation.

-          The manuscript is too long due to the unnecessary explanation. For example, sentences line 389-390 should be removed and shown in figure legends.

-          Discussion in this manuscript must be rewritten to explain more straightforward and scientifically.

Round 2

Reviewer 1 Report

The revised manuscript is greatly improved and answered all questions. It could be published in this version.

Reviewer 2 Report

The authors addressed the issue well and rewrote the manuscript in the more scientific way. I appreciate your effort. I recommend this manuscript to publish